# Evaluation of the insulin-dependent and -independent hypoglycemic effects and understanding their breakdown in the progression of obesity using mice

Fusako Kojima[1,2], So Morishita[1], Shinsuke Uda[1,3], Noriko Yutsudo[1], Yoshihiro Ogawa[2], Hiroyuki Kubota[1]*

1 Division of Integrated Omics, Medical Research Center for High Depth Omics, Medical Institute of Bioregulation, Kyushu University, Fukuoka, Japan, 2 Department of Medicine and Bioregulatory Science, Graduate School of Medical Sciences, Kyushu University, Fukuoka, Japan, 3 Center for Information and Data Science Education, Yamaguchi University, Yamaguchi, Japan

* kubota.hiroyuki.855@m.kyushu-u.ac.jp

## Abstract

Temporal changes in blood glucose levels are useful in assessing glucose tolerance. While blood glucose levels are known to be tightly regulated by insulin-dependent and -independent (glucose effectiveness) hypoglycemic effects, their temporal changes and the organs involved have not been sufficiently explored. This study showed how the relative strengths of these effects change over time during the glucose clamp test, using biological experiments and a mathematical model. We found that the glucose effectiveness exhibited a transient effect within 60 min, whereas the insulin-dependent effect showed a sustained effect intensified after 60 min. Additionally, we found that the liver and muscles were responsible for glucose effectiveness and insulin-dependent hypoglycemic effects, respectively. We applied this model to obese mice and found that both hypoglycemic effects decreased as obesity progressed. These results provide insights into hypoglycemic effects and have important implications for the assessment and interpretation of glucose tolerance in mice.

## Introduction

Glucose tolerance tests, such as the intravenous glucose tolerance test (IVGTT), oral glucose tolerance test (OGTT), insulin tolerance test (ITT), and hyperglycemic clamp test, are indispensable for evaluating both deterioration and improvements in individuals with insulin sensitivity and glucose intolerance. However, due to mutual feedback between blood insulin and glucose, which gives rise to complex responses (e.g., insulin secretion, glucose uptake, gluconeogenesis, etc.), identifying those responses that are improved or those that are worsened is a challenge. Glucose effectiveness is the ability of glucose to normalize its concentration [1], promote its

**Data availability statement:** All relevant data are within the paper and its Supporting Information files.

**Funding:** The Japan Society for the Promotion of Science KAKENHI, grant number JP20H03237. the Japan Science and Technology Agency (JST) Moonshot R&D, grant number JPMJMS2022-8.

**Competing interests:** The authors have declared that no competing interests exist.

uptake into the body, and suppress endogenous glucose production (EGP) [2–4]. For more than 80 years, the insulin-independent effect has been reported [5,6]. The mass-action effect of glucose itself and its diffusion into the interstitial fluid have been proposed as potential mechanisms underlying glucose effectiveness [2,7]. However, its characteristics underlying glucose effectiveness, including temporal changes and the specific sites of glucose absorption, remain poorly understood. In the 1970s, Bergman et al. named the insulin-independent effect "glucose effectiveness" [8], which is calculated using a mathematical model. Previous studies have suggested that glucose effectiveness plays an important role in the development of impaired glucose tolerance and is an important determinant of the progression of type 2 diabetes mellitus (T2DM) [9,10]. However, its role in patients with T2DM is controversial, and little research has been conducted on its detailed mechanism [2].

Many mathematical models have been developed since the 1960s to quantitatively evaluate the mechanisms of blood glucose regulation. In 1961, Bolie et al. proposed a mathematical model of the glucose-insulin relationship using ordinary differential equations [11]. In 1979, Bergman et al. proposed a minimal model of glucose-insulin dynamics [8]; this model enables the quantitative evaluation of insulin sensitivity and glucose effectiveness and has been used in various studies. Since then, several models, including the modified Bergman's minimal model, have been developed to express the quantitative relationship between blood glucose and insulin levels [12–21]. It has been reported that glucose effectiveness is responsible for approximately 45–65% of glucose disposal in humans, based on the results of a minimal model analysis using IVGTT and glucose clamp experiments [7,22]. Furthermore, in another study, it was found to potentially exceed the insulin-dependent hypoglycemic effect in an IVGTT in rats [23]. These results suggest that glucose effectiveness is a major factor in the maintenance of blood glucose levels.

In diabetes research conducted in mice, glucose tolerance, including insulin resistance, insulin secretion ability, and drug administration, has been evaluated in both diet-induced obese and genetically modified mice using the abovementioned mathematical models. Glucose effectiveness has also been evaluated in some cases [24–32]. However, most animal studies have been conducted using indices centered on insulin resistance and insulin secretion ability. In fact, glucose effectiveness has rarely been considered. This may be because the detailed mechanism of glucose effectiveness is not clear and the mathematical models have not shown consistent results related to glucose effectiveness [2]. Thus, the evaluation of glucose effectiveness on the progression of glucose intolerance remains controversial. Reproducing differences in the dynamics of the two hypoglycemic effects will help to advance our understanding of the regulatory mechanisms between blood glucose and insulin, as well as the mechanisms involved in the progression of T2DM. Moreover, a better understanding of the mechanisms underlying the improvement/deterioration of glucose metabolism will provide new insights useful for both treatment and drug discovery.

In this study, we sought to clarify the magnitude of glucose effectiveness on hypoglycemic effects and examined temporal dynamic changes in insulin-dependent

hypoglycemic effects and glucose effectiveness after glucose loading using biological experiments and a mathematical model. Additionally, we conducted hyperglycemic clamp studies in obese mice of various ages and, using a mathematical model, elucidated the changes in parameters associated with the progression of obesity.

## Results

### Blood glucose levels decreased even without insulin secretion in the IVGTT

To confirm the relationship between insulin secretion and its hypoglycemic effects, we conducted an IVGTT with or without somatostatin. Blood glucose levels decreased after reaching a peak even when insulin secretion was suppressed (S1 Fig). As shown in S1B Fig, blood insulin showed a biphasic response in mice without somatostatin: a transient first-phase insulin secretion followed by sustained second-phase secretion [33]. In contrast, insulin secretion was suppressed in somatostatin-treated mice. This indicates that glucose levels are mainly regulated in an insulin-independent manner, at least during the 90 min of the IVGTT. These results are consistent with previous studies [34,35]. However, somatostatin inhibits not only insulin secretion but also glucagon secretion in pancreatic islet cells. Thus, the similar time courses of blood glucose levels observed in IVGTT with or without somatostatin (S1 Fig) may result from abnormal glucose regulation due to suppressed glucagon secretion. Therefore, all subsequent experiments were conducted without somatostatin.

### First-phase insulin secretion has no significant effect on hypoglycemic effects

We further examined the effects of the first-phase insulin secretion on hypoglycemia. The first-phase insulin secretion is observed only when the β-cells are exposed to rapid changes in glucose concentrations [36]. Therefore, we used a hyperglycemic clamp with slow and fast initial infusion rates (Fig 1A). To clarify the effects of the second-phase insulin secretion, the clamp test period was extended to 3 h. At a slow initial administration rate, blood glucose levels gradually increased, and the first-phase insulin secretion was not observed (Fig 1B and 1C). At a fast initial administration rate, blood glucose levels rapidly increased and blood insulin levels showed a transient response, indicating first-phase insulin secretion (Fig 1E and 1F). The area under the curves (AUCs) of insulin from 0 to 10 min and after 10 min are often used as indices of first- and second-phase insulin secretion in the hyperglycemic clamp and IVGTT, respectively. In this experiment, the insulin AUC at 0–10 min of the fast rate was significantly larger than that at the slow rate ($p=0.002253$, Welch's $t$ test, Fig 1H); however, there was no significant difference in the insulin AUC at 10–180 min between them ($p=0.9924$, Welch's $t$ test, Fig 1I). At both rates, interestingly, the amounts of infused glucose for maintaining the target blood glucose level decreased after the initial glucose infusion (≤ 20 min) and then continued to increase after about 60 min (Fig 1D and 1G). These results clearly indicate that there are two hypoglycemic effects regardless of the glucose infusion rate. A comparison of the total infused glucose amounts between the two groups during the clamp tests showed no statistically significant difference ($p=0.5969$, Welch's $t$ test, Fig 1J). These results indicate that the first-phase insulin secretion has little hypoglycemic effect compared to glucose effectiveness.

### Glucose is time-dependently taken up in an insulin-independent and -dependent manner

To further examine the relationship between blood insulin levels and glucose uptake, we divided the insulin time course into three parts and assessed the correlation between the insulin AUC and the amount of infused glucose (see Methods). In both groups, there was no correlation between insulin AUC and the amount of infused glucose from 0 to 60 min (Fig 2A-C). However, in the group with a slow rate of initial glucose administration, a weak positive correlation was observed at 60–120 min, and a strong positive correlation was observed between 120–180 min (Fig 2B). In the group with a fast rate of initial glucose administration, there were strong positive correlations between 60 and 120 and 120–180 min (Fig 2C). These results indicate that, regardless of the presence or absence of the first-phase insulin secretion, glucose uptake during the first 60 min of the hyperglycemic clamp was mainly caused by an insulin-independent effect (glucose effectiveness), and

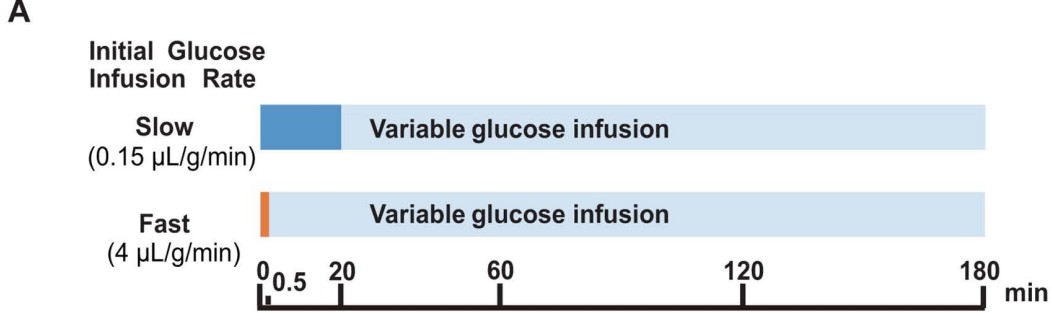

**Fig 1. Depending on the glucose infusion rates, the magnitude of first-phase insulin secretion differed, but did not affect the total glucose uptake in the hyperglycemic clamp test.** Hyperglycemic clamp test was performed in 14-week-old chow-fed mice at slow or fast initial administration rates. (A) Schematic view of the hyperglycemic clamp test. Blood glucose (B and E) and insulin levels (C and F), and the amount of infused glucose (D and G) during hyperglycemic clamp with slow-rate ($n$ = 10, 0.15 μL/g/min) (B-D) or fast-rate ($n$ = 10, 4 μL/g/min) (E-G) initial glucose administration. Insulin AUC 0-10 min (H), Insulin AUC 10-180 min (I), and the total amount of infused glucose in each group (J) during the hyperglycemic clamp. All results are expressed as mean ± SE ($n$ = 10); ** $p$ < 0.01, Welch's $t$ test.

insulin had a larger effect after 60 min. The amount of infused glucose for the first 60 min, when glucose effectiveness dominated, was comparable to that for the last 60 min when insulin-dependent effects dominated (Fig 2D). This indicated that glucose effectiveness contributed to glucose uptake to the same extent as the insulin-dependent hypoglycemic effect in the hyperglycemic clamp in mice. This explains why bolus stimulation with or without somatostatin by IVGTT showed a similar time course owing to glucose effectiveness (S1 Fig).

### Glucose uptake in the liver and muscles showed different time-dependent characteristics

We investigated the organs in which glucose is taken up by insulin-independent and insulin-dependent mechanisms. Diffusion into the interstitial fluid is a potential mechanism underlying glucose effectiveness. However, since the diffusion of glucose throughout the body would also increase blood glucose levels, the diffusion alone is insufficient to fully account for the insulin-independent decreases in blood glucose levels to the basal level (S1 Fig). Therefore, we focused on the liver and muscles, which play major roles in glucose uptake and storage [37]. We examined the glucose uptake by the liver and gastrocnemius muscles (muscles) of mice at 0, 10, 20, 60, 120, and 180 min during a hyperglycemic clamp with 2-DG (Fig 3 and S2 Fig). The amount of glucose taken up by the liver was significantly higher than that taken up by the muscles at 10 and 20 min ($p$ = 0.000735 and 0.04836, respectively, Welch's $t$ test, Fig 3C). The time courses of blood glucose and insulin levels, and infused glucose were comparable with or without 2-DG (Fig 1B-D, and 3B). Glucose uptake in both organs increased after 60 min, and the uptake at 180 min was similar. Glucose uptake in the liver showed a biphasic pattern, whereas that in the muscles showed a monophasic pattern (Fig 3C). These results indicate that the liver is related to both insulin-independent and insulin-dependent glucose uptake and that the muscles are related to insulin-dependent glucose uptake.

### Developing a new mathematical model that can reproduce glucose uptake in addition to blood glucose and insulin levels

Next, we examined the relationship between blood glucose and insulin levels using a mathematical model. To evaluate the time-dependent glucose uptake, the amount of glucose uptake was considered as an evaluation function, in addition to blood glucose and insulin levels. First, we developed a model using our previous one [38]; however, it could not reproduce the data (the model reproduced blood glucose and insulin levels, but not the amount of glucose uptake). This result was reasonable because, under our experimental conditions in which transient insulin increase was not observed, all previous models, including ours, could not reproduce the transient glucose uptake. This is because glucose uptake depends on blood glucose levels and essentially follows the same pattern as blood glucose. Therefore, we developed a new model based on the previous model (Fig 4A, see Methods). Because glucose effectiveness is considered a transient glucose uptake based on the above experiments (Fig 2 and 3), $stG$ was added as a temporary pooling site for glucose. The *flux* 5, which is the influx of glucose into $stG$ depends on the difference in $stG$ from the upper limit of the pool ($k_7$), that is, on the remaining pool volume of $stG$ and the blood glucose level, but not on insulin. Owing to this characteristic, *flux* 5 is large at the start of the glucose clamp because $stG$ is not yet filled, but decreases once $stG$ becomes filled. This behavior accounts for transient glucose uptake. The *flux* 6, represents the consumption of pooled glucose in $stG$ depending on blood insulin levels. The *flux* 1 represents EGP. EGP is primarily

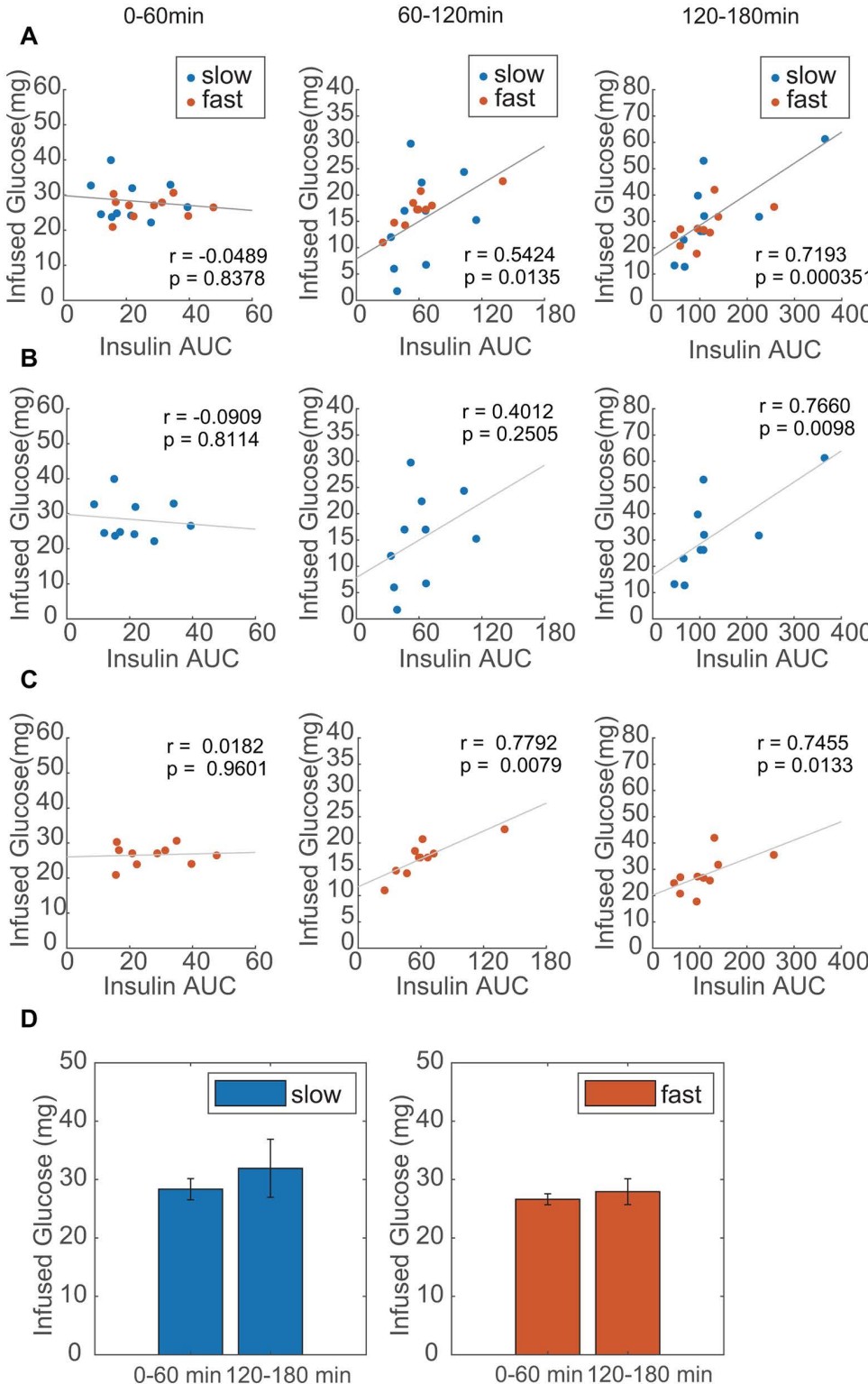

**Fig 2. Time-dependent correlations between blood insulin levels and the amount of infused glucose during the hyperglycemic clamp test.** (A-C): Spearman's rank correlation between blood insulin levels and the amount of infused glucose for each 60 min during the hyperglycemic clamp test for all results (A), slow-rate ($n = 10$, 0.15 μL/g/min) (B), and fast-rate ($n = 10$, 4 μL/g/min) initial glucose administration (C). Blue and orange circles represent

the results with slow-rate ($n = 10$, 0.15 μL/g/min) and fast-rate ($n = 10$, 4 μL/g/min) initial glucose administration, respectively. (D) Comparison between infused glucose amounts for the first 60 min and the last 60 min in the hyperglycemic clamp. Blue and orange bars indicate data with slow and fast initial infusion rates, respectively ($n = 10$).

activated by glucagon, but it is generally recognized that blood insulin and glucagon levels are inversely correlated. Moreover, because EGP has been reported to be inhibited by blood glucose levels via glucose effectiveness [39], *flux* 1 is assumed to be suppressed by both blood insulin and glucose levels. The parameter $k_8$ defines the relative contribution of insulin and glucose to the suppression of EGP.

Using this model, we reproduced blood glucose and insulin levels and the amount of glucose uptake in 10 control 14-week-old mice (Fig 4B and S3 Fig). We confirmed the behaviors of *flux* 5 and *flux* 2, which correspond to glucose effectiveness and insulin-dependent glucose uptake, respectively. In Fig 4C, *flux* 5 and *flux* 2 were represented by an integrated value of 10 min at 10, 20, 60, 120, and 180 min, which is comparable to the estimated amount of glucose uptake by the liver and muscles (Fig 3C). The amount of glucose taken up into *stG* through *flux* 5 rapidly increased, and at approximately 60 min, *stG* almost reached a plateau (Fig 4C). Therefore, *flux* 5 showed the largest values at the beginning and decreased after 60 min, whereas *flux* 2 gradually increased monotonically from beginning to end. These results indicate that our model reproduced time-dependent glucose uptake: glucose effectiveness was dominant in the first 60 min, and insulin-dependent glucose uptake became dominant after 60 min. These results were consistent with those of glucose uptake by the liver and muscles in the hyperglycemic clamp with 2-DG (Fig 3C). Thus, our model captured the characteristics of time-dependent glucose uptake caused by glucose effectiveness and insulin-dependent hypoglycemic effects in control mice.

### Evaluation of the changes in the parameters during the progression of obesity

As previously described, there is a feedback loop between blood insulin and glucose that plays an important role in glucose homeostasis. Obesity often causes impaired responses, including feedback, which leads to glucose intolerance. Mathematical models have been developed to evaluate which reaction(s) is impaired. However, the previous models may not precisely evaluate the responses because they do not reproduce the time-dependent glucose uptake. Therefore, we attempted to infer changes during the progression of obesity by comparing the parameters using our model. We used 10, 14, 18, 28, and 36-week-old HFD-fed mice for the experiments and performed a hyperglycemic clamp with the same experimental condition (slow initial administration rate). Body weight increased in a time-dependent manner and fasting blood glucose and insulin levels were significantly higher than those in chow-fed mice, indicating glucose intolerance (Fig 5A-C). In the hyperglycemic clamp, we set the target blood glucose levels at 200 mg/dL higher than the fasting condition to ensure identical changes in blood glucose levels and maintain a constant blood glucose level (Fig 5D). The pattern of glucose administration over time during the hyperglycemic clamp in 10−18 weeks HFD-fed mice was similar to that in 14-week-old chow-fed mice; the amount of administered glucose decreased after 20 min of the initial infusion and then increased in a time-dependent manner from approximately 60 min (Fig 5F). In contrast, in 28- and 36-week-old HFD-fed mice, the amount of administered glucose did not decrease compared to the other groups; more glucose was required to maintain the target blood glucose levels, and the time courses did not show time-dependent increases (Fig 5F and 5H). Blood insulin levels during the hyperglycemic clamp increased in a time-dependent manner in all mice, and these levels also increased as the age of the mice increased (Fig 5E and 5G). We developed models for individual mice that reproduced blood glucose and insulin levels, and the amount of glucose uptake (Fig 6 and S3 Fig). The estimated parameters were compared (Fig 7 and S1 Table). The statistical significance indicated by asterisks in Fig 7A was determined using Welch's *t* test with Benjamini–Hochberg correction (\*$p < 0.05$, \*\*$p < 0.01$, \*\*\* $p < 0.001$, \*\*\*\* $p < 0.0001$). Parameters showing differences among mouse groups ($k_2$, $k_3$, $k_5$, and $k_7$) were considered to exhibit age-dependent monotonic trends.

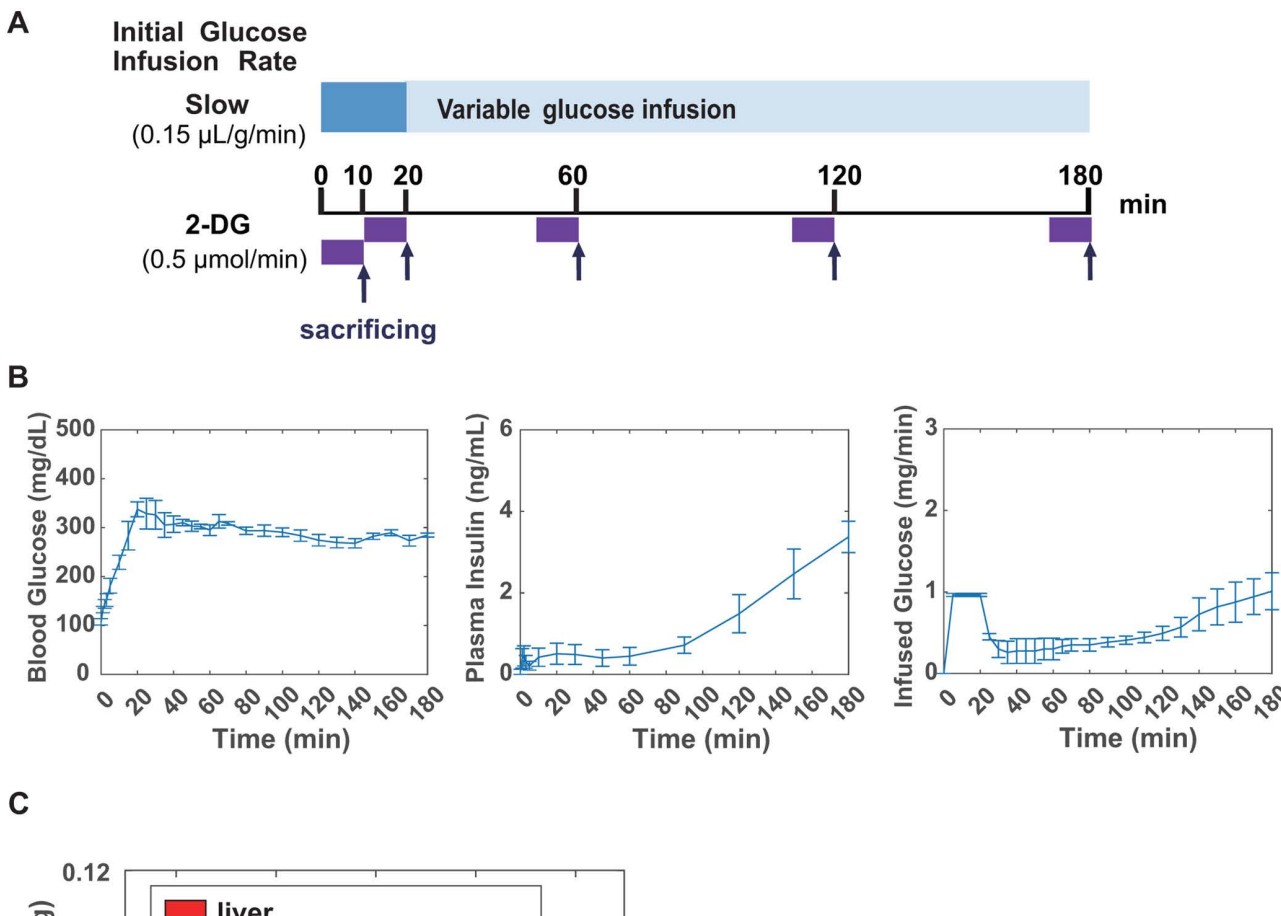

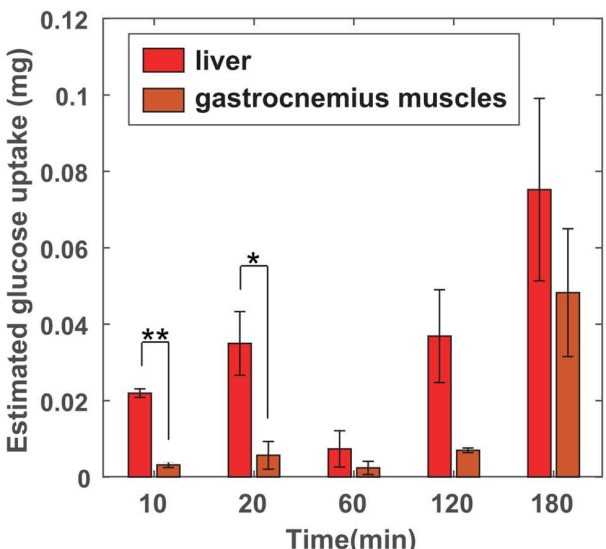

**Fig 3. Time-dependent glucose uptake in the liver and muscles during the hyperglycemic clamp test.** (A) Schematic view of the hyperglycemic clamp test with 2-DG. (B) The time courses of blood glucose (left) and insulin (middle) levels, and infused glucose (right) during the hyperglycemic clamp with 2-DG for 180 minutes. All results are expressed as mean±SE ($n=3$). (C) Red and orange bars indicate the amount of 2-DG taken up by the liver and muscles, respectively. Data are means±SE ($n=3$ at 10, 20, 60, 120,180 min); *$p<0.05$, ** $p<0.01$, Welch's $t$ test.

*Equations in panel A:*

$$flux\ 1 = k_1 / (1 + I + k_8 * G)$$
$$flux\ 2 = k_2 * G * I$$
$$flux\ 3 = k_3 * G$$
$$flux\ 4 = k_4 * I$$
$$flux\ 5 = k_5 * G * (k_7 - stG)$$
$$flux\ 6 = k_6 * stG * I$$
$$influx\ G = f(t)$$

$G$ : Blood glucose
$I$ : Blood insulin
$stG$ : Storage of glucose due to glucose effectiveness

**Fig 4. Reproduction of the experimental data by the developed model.** (A) Schematic view of the model. (B) The time courses of blood glucose (left) and insulin (middle) levels, and the amount of glucose uptake (right) during the hyperglycemic clamp test in 14-week-old mice fed a normal chow diet (CD). Orange dots and blue lines indicate experimental and simulation results, respectively. (C) The integrated values during 10 minutes of *flux* 5 and *flux* 2 at 10, 20, 60, 120, and 180 min, and the simulation time course of *stG*. "CD 14wk #" in the figure labels denotes individual mouse IDs.

Therefore, we next examined whether these parameters showed age-dependent monotonic trends using Williams' test. Parameter $k_2$, representing insulin-dependent glucose uptake, showed significant decreases in HFD36wk ($p < 0.001$), HFD28wk ($p < 0.001$), and HFD18wk ($p = 0.015$) mice relative to CD14wk mice. The HFD14wk ($p = 0.0568$) and HFD10wk ($p = 0.1177$) mice showed downward trends, though these did not reach statistical significance. These findings suggest

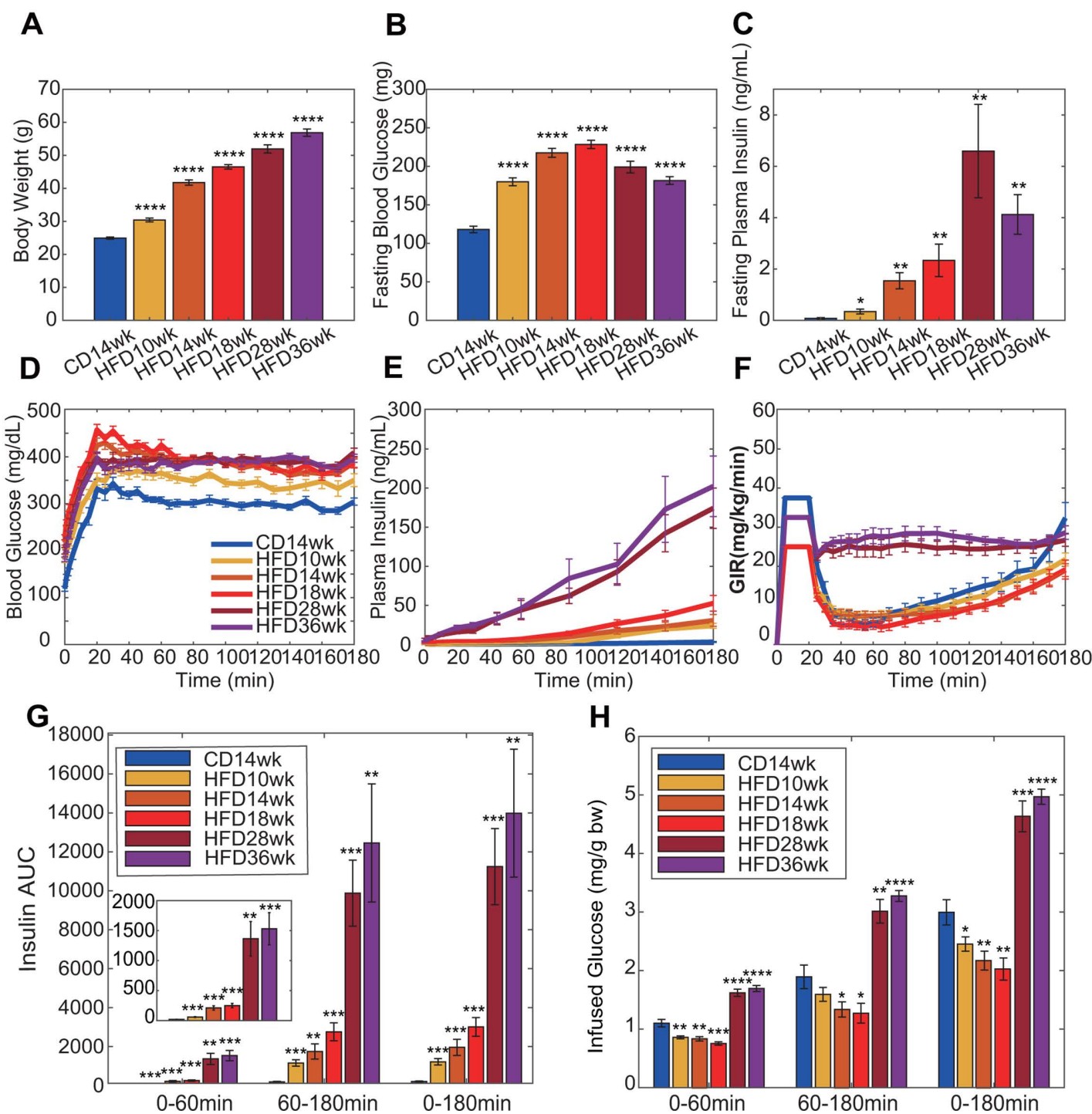

**Fig 5. The results of the hyperglycemic clamp in 14-week-old chow-fed mice and 10, 14, 18, 28, and 36-week-old HFD-fed mice (*n* = 13, 13, 16, 13, 10, and 10).** (A-C): Bar graphs of body weights (A), fasting blood glucose (B), and insulin levels (C) in each condition of mice. (D-F): Time courses of blood glucose (D) and insulin levels (E), and glucose infusion rate (F) in each condition of mice during the hyperglycemic clamp test. (G, H): Bar graphs indicate insulin AUC (G) and infused glucose amounts (H) for the first 60 min, 60-180 min, and 0-180 min in each condition of mice. Data are means ± SE. Welch's *t* test with Benjamini-Hochberg *p*-value adjustment (**p* < 0.05, ***p* < 0.01, *** *p* < 0.001, **** *p* < 0.0001).

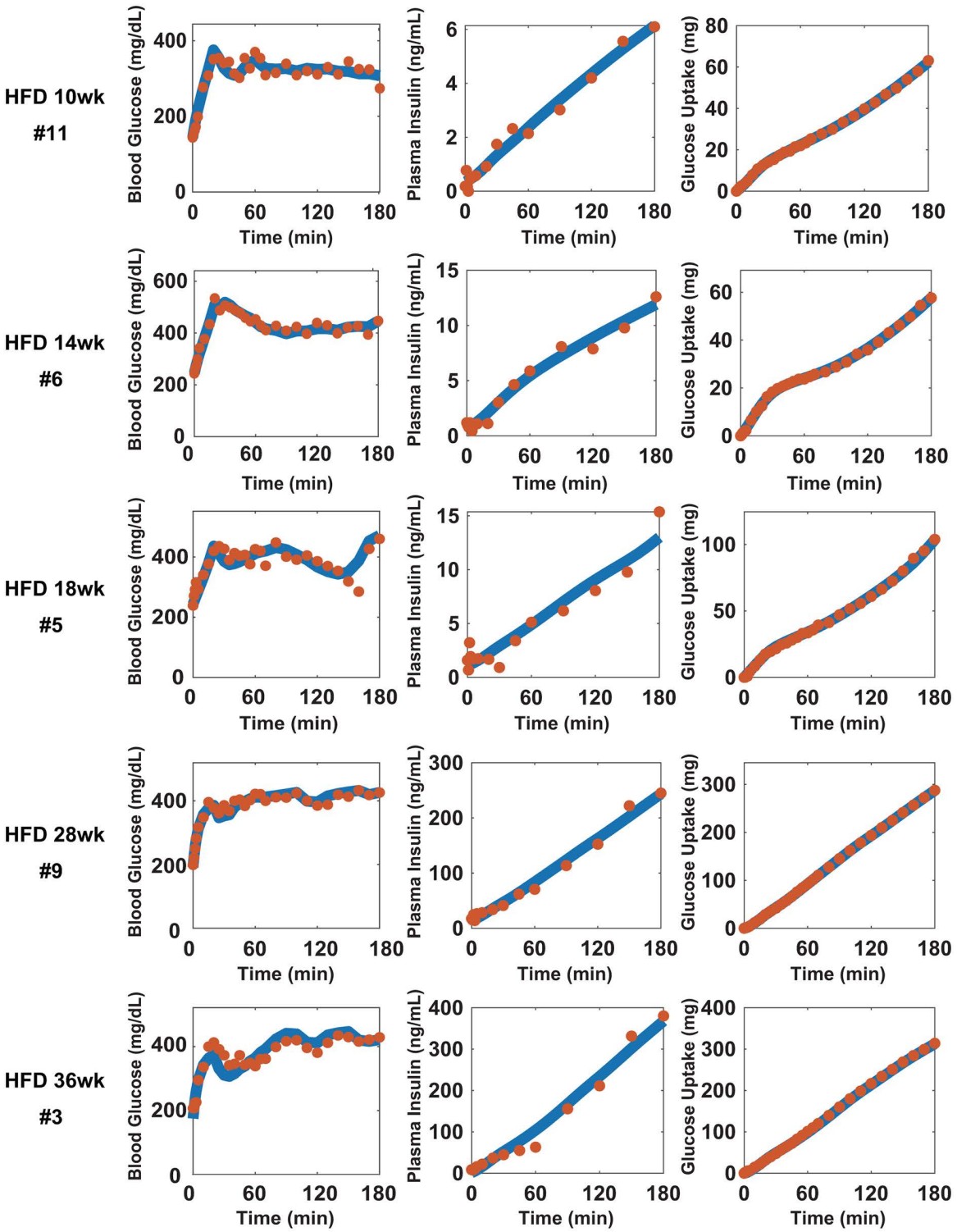

**Fig 6. Experimental results versus mathematical model simulations.** The representative time courses of blood glucose (left) and insulin (middle) levels, and the total amount of glucose uptake (right) during the hyperglycemic clamp in indicated week-old HFD-fed mice. Orange dots and blue lines indicate experimental and simulation results, respectively.

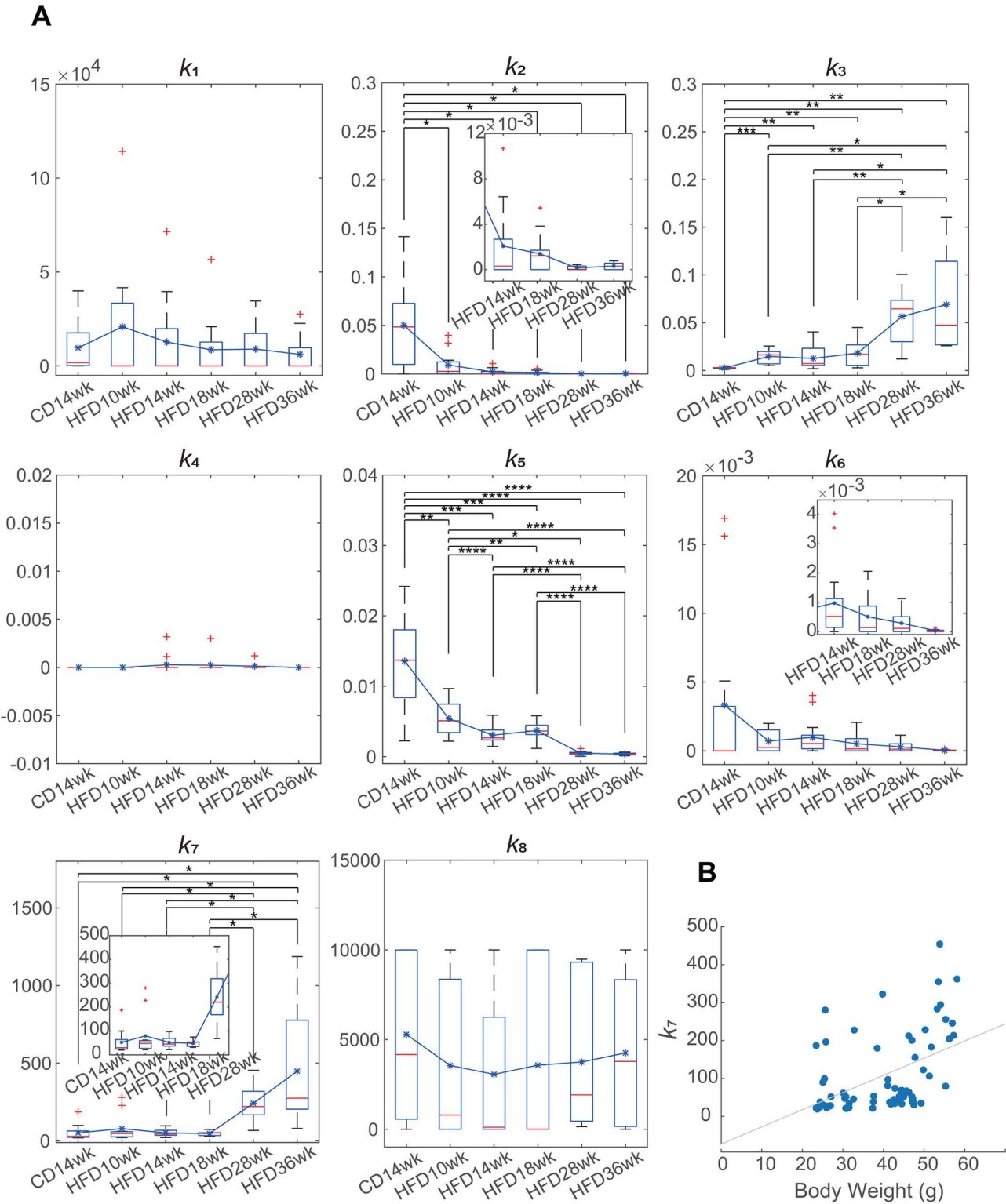

**Fig 7. Parameter alterations in the progression of obesity.** (A) Comparison of the parameters among the 14-week-old chow-fed mice and 10, 14, 18, 28, and 36-week-old HFD-fed mice ($n$ = 13, 13, 16, 13, 10, and 10). The results were shown by the box plots. Blue line graphs indicate the mean values of each condition. Welch's $t$ test with Benjamini-Hochberg $p$-value adjustment was used to compare (*$p$ < 0.05, **$p$ < 0.01, *** $p$ < 0.001, **** $p$ < 0.0001). (B) Spearman's rank correlation between the parameters of $k7$ and the body weights of all of the mice ($n$ = 75, $r$ = 0.6534, $p$ = 2.0963e-10).

that insulin sensitivity moderately decreased during obesity progression under HFD feeding. Parameter $k_5$, representing glucose effectiveness, showed an age-dependent decreasing trend ($p < 0.001$ for each hypothesis test). This result indicates a monotonic decreasing trend in glucose effectiveness with increasing age under HFD feeding, suggesting an age-dependent impairment of insulin-independent glucose uptake in obese mice. Parameter $k_3$, representing insulin secretion, showed an age-dependent increasing trend ($p < 0.001$ for each hypothesis test), indicating a progressive enhancement of insulin secretion during obesity progression under HFD feeding. The increase in $k_3$ in HFD28wk and HFD36wk mice suggests that the increase in infused glucose (Fig 5F) was not due to an improvement in insulin sensitivity but increased insulin secretion. Moreover, the decrease in $k_2$ indicates that insulin-dependent glucose uptake was down-regulated in these mice. These results together indicate that insulin resistance further progressed. Parameter $k_7$, representing the capacity of $stG$, showed a monotonic increasing trend (HFD36wk, $p < 0.001$; HFD28wk, $p < 0.001$; HFD18wk, $p = 0.00807$). In addition, since increasing trends were not observed in younger mice (HFD14wk, $p = 0.06645$; HFD10wk, $p = 0.10421$), these results suggest that glucose storage capacity increases some time after the onset of obesity. Therefore, we examined the correlation between $k_7$ and body weight and found a positive correlation ($r = 0.6534$, $p = 2.0963\text{e-}10$, Spearman's rank correlation analysis, Fig 7B). These results indicate that our model could reflect changes in glucose homeostasis during the progression of obesity (see Discussion).

## Discussion

In this study, we focused on temporal changes in glucose uptake and the sites of glucose absorption during the glucose clamp test. It has been reported that glucose uptake consists of both insulin-dependent and -independent responses (the latter referred to as glucose effectiveness). Glucose effectiveness is defined as the ability of glucose itself to suppress EGP and enhance glucose uptake, a process that is insulin-independent [39,40]. Based on our results, glucose effectiveness may be attributed to glucose diffusion into the interstitial fluid and uptake by GLUT2-expressing organs (see below), both of which are insulin-independent. However, considering that blood glucose levels insulin-independently decreased to the basal level (S1 Fig) in an IVGTT, glucose effectiveness cannot be explained solely by glucose diffusion. Furthermore, suppression of EGP by glucose effectiveness [2] cannot be accounted for by glucose diffusion alone. These results indicate that the liver is related to glucose effectiveness and are consistent with the previous hypothesis [7]. While we do not exclude the contribution of diffusion into the interstitial fluid to transient glucose storage, our conclusions would not be altered.

We examined glucose effectiveness and insulin-dependent hypoglycemic effects using biological experiments and a mathematical model in an attempt to reveal alterations in the responses in the progression of obesity. We found that glucose effectiveness primarily takes effect within 60 min of glucose administration, while insulin-dependent hypoglycemic effects are predominantly enhanced from 60 to 180 min in the hyperglycemic clamp. Furthermore, the hypoglycemic effects in the first and last 60 min were approximately the same, indicating that the glucose effectiveness was comparable to the insulin-dependent effect. This is in agreement with previous findings that insulin-dependent and -independent glucose uptake is approximately 50% in healthy humans and dogs [7,39–41]. Moreover, the notion that insulin-independent mechanisms predominantly contribute to the initial decrease in blood glucose levels, whereas insulin action is delayed and progressively increases over time, is also supported by several previous studies. For example, Quon et al. [42] performed an IVGTT in patients with type 1 diabetes mellitus, administering insulin in a manner that mimicked endogenous secretion patterns. Their analysis demonstrated that glucose clearance was primarily insulin-independent for approximately 20 minutes after administration, after which it accelerated due to insulin effects. Furthermore, Mossberg et al. [35] reported that in a hyperinsulinemic-euglycemic clamp test with fluorodeoxyglucose in rabbits, although not statistically significant, there were two distinct peaks in glucose uptake. Factorial analysis also indicated that the effect of insulin increased in a time-dependent manner. These results are also consistent with those shown in S1 Fig, although the contribution of somatostatin cannot yet be excluded: the initial decrease in blood glucose levels is attributed to insulin-independent uptake,

and by the time insulin action becomes significant, blood glucose levels have already decreased. Therefore, no difference might be observed between the conditions with or without somatostatin. However, these results differ markedly from the dynamics predicted by mathematical models, such as Bergman's minimal model. This discrepancy arises because such models cannot reproduce transient glucose uptake under the conditions of this experiment, where blood glucose levels are constant and there is no transient insulin response (first-phase insulin secretion). Indeed, Cobelli et al. [43] applied a minimal model to analyze Quon's data obtained by administering insulin to patients with type 1 diabetes mellitus in a manner that mimicked endogenous secretion patterns. Their analysis suggested the existence of additional compartments required to reproduce early glucose uptake.

Several studies have used individual indices and their combinations to evaluate glucose effectiveness in humans and animals. Our study highlights the possibility of reconsidering these interpretations. For example, indices such as the insulin/glucose AUC ratio and Matsuda-DeFronzo Index, which are derived from blood glucose levels during glucose tolerance tests, have traditionally been interpreted as insulin sensitivity [44–46]. However, they may actually primarily assess glucose effectiveness or may not sufficiently differentiate between insulin-dependent effects and glucose effectiveness. Our study demonstrated the importance of considering the time elapsed after glucose loading to distinguish between these two effects.

We also experimentally examined glucose uptake using 2-DG and found that the liver and muscles showed bimodal and monotonically increasing responses, respectively. To the best of our knowledge, this is the first study to demonstrate time-dependent insulin-dependent and -independent glucose uptakes and their involved organs. It can be considered that glucose is also diffused into the interstitial fluid. However, glucose uptake by the liver within the first 60 min (glucose effectiveness) was comparable to that observed after 60 min (insulin-dependent glucose uptake) (Fig 3C), indicating a sufficient amount of glucose was insulin-independently taken up by the liver. Thus, even though the interstitial fluid contributed to glucose effectiveness, the liver also contributed comparably to insulin-dependent glucose uptake. Glucose transporter 2 (GLUT2) and glucose transporter 4 (GLUT4) take up glucose in an insulin-independent and -dependent manner, respectively, and are predominantly expressed in the liver and muscle. Therefore, we hypothesized that the organs in which GLUT2 and GLUT4 are predominantly expressed may be responsible for glucose effectiveness and insulin-dependent hypoglycemic effects, respectively.

We compared the temporal changes in glucose uptake over time in the liver and muscles, which are potential representative organs for glucose uptake, with the temporal changes in the two hypoglycemic effects inferred from the mathematical models, *flux* 5 and *flux* 2. In our simulation, *flux* 5, which represents insulin-independent glucose uptake towards *stG* (transient storage of glucose), increased rapidly immediately after glucose administration, and *stG* almost reached a plateau at approximately 60 min (Fig 4C). This may indicate that GLUT2, which takes up glucose depending on the glucose concentration, cannot transport glucose any further after reaching the same glucose concentration inside and outside the blood vessels. Based on our hypothesis, glucose is transiently stored in organs where GLUT2 is primarily expressed, such as the liver. Similar to *flux* 5, the amount of 2-DG uptake in the liver during the hyperglycemic clamp showed a transient increase within 60 min (Fig 3C). Our results and hypothesis are consistent with the previous study [7]. Note that since *stG* represents a transient storage of glucose, it also contains the effects of glucose diffusion into the interstitial fluid. In addition, our results showed a positive correlation between mouse body weight, which is considered to correlate with liver weight, and $k_7$, which represents the capacity of *stG* (Fig 7B). These results support the hypothesis that glucose effectiveness takes effect in the liver. After transient glucose uptake, both *flux* 5 in the simulation and 2-DG uptake in the liver during the hyperglycemic clamp started to increase again 60 min after glucose administration, when *stG* almost reached a plateau. This indicates that the activation of intracellular glycolysis and glycogen synthesis in response to increasing insulin concentrations promotes GLUT2-mediated glucose uptake. This is represented by *flux* 6, in our model. In contrast, *flux* 2, which represents insulin-dependent glucose uptake, increased consistently and monotonically. Based on our hypothesis, this response is observed in organs that express Glut4, such as muscle. This is consistent with the time course of

2-DG uptake in the muscles. Insulin-dependent glucose uptake increased from 60 to 180 min after glucose administration, indicating that insulin-dependent glucose uptake in the liver and muscles was enhanced instead of glucose effectiveness, which was less effective 60 min after glucose administration. Sjöstrand et al. reported that during a hyperinsulinemic-euglycemic clamp test, the rise in insulin levels in the muscle interstitial fluid and the onset of insulin action in the muscle are delayed [34]. This delay in transcapillary insulin delivery may be one factor contributing to the delayed organ-selective insulin action in our study. By combining biological experiments and mathematical model analyses, it was strongly suggested that mice respond to a glucose load by symphonizing two hypoglycemic effects with different durations of action.

In addition, we examined obesity in mice of different ages using this model. As obesity progressed, insulin secretion ($k_3$) increased, while insulin sensitivity ($k_2$) decreased (Fig 7A). Similar to previous models, our model captured changes in insulin sensitivity ($k_2$) and secretion ($k_3$). Parameters $k_5$ and $k_7$, which are related to glucose effectiveness, also changed with the progression of obesity. Glucose effectiveness ($k_5$) decreased, whereas the maximum capacity for glucose storage via glucose effectiveness ($k_7$) increased as obesity progresses. Hepatic glucose uptake is mainly regulated by glucokinase (GK) [47], and its capacity ($stG$) appears to be related to the liver size and conditions. Hepatic glucose uptake is impaired in individuals with obesity and T2DM [48], and impaired GK is one of the causes [48,49]. GK is controlled by glucokinase regulatory protein (GKRP) and is activated by the glucose-dependent dissociation of GK and GKRP. Watanabe et al. reported that $NAD^+$-dependent Sirt2 facilitates the dissociation of GK and GKRP in mice [50], and that a high-fat diet reduces hepatic $NAD^+$ levels, thereby decreasing hepatic Sirt2 activity. They further demonstrated that glucose-dependent glucose uptake was enhanced by Sirt2 in an insulin-independent manner. Therefore, decreased GK activity in the liver may play a role in glucose effectiveness. Further experimentation and verification are required to elucidate the detailed mechanisms. This study provides insights useful for further investigation.

A mathematical model that distinguishes and assesses glucose effectiveness from insulin-dependent hypoglycemic effects can help identify the molecules and factors that affect glucose effectiveness. This may lead to the development of hypoglycemic medications with new mechanisms and increase the options for early intervention in people with prediabetes and diabetes. In conventional glucose tolerance assessments based on glucose tolerance tests, it is unclear which hypoglycemic effect occurs, and when it takes effect after glucose loading. The question of when insulin-independent and insulin-dependent hypoglycemic effects occur after glucose loading provides new insights into the currently used measures for assessing glucose tolerance. Various glucose tolerance tests have been performed in obese genetically engineered mice to evaluate glucose intolerance and the effects of hypoglycemic medication. In the case of OGTT and IVGTT studies in animal experiments, glucose tolerance is often evaluated and compared based on the AUC of glucose and insulin. However, when evaluating the results within 60 min of glucose administration, they may mainly reflect glucose effectiveness. Our findings indicate that time-dependent glucose effectiveness and insulin-dependent hypoglycemic effects should be considered in any glucose tolerance tests to interpret results more accurately, even if the experimental conditions are different. Our model is expected to provide qualitatively different information compared to previous models, and may also provide insight into the mechanisms of impaired glucose tolerance due to other non-diabetic diseases, such as hepatic cirrhosis. In our model, EGP is represented by *flux* 1 and is inhibited by insulin and glucose levels. It is also known that insulin and glucose suppress glucagon secretion. Since EGP is primarily regulated by glucagon, the representation of *flux* 1 is consistent with these studies. However, multiple factors affecting EGP, such as epinephrine, cortisol, growth hormone, and free fatty acids, have been reported. Therefore, incorporating these additional factors into our model is expected to provide a more comprehensive understanding of blood glucose homeostasis.

In humans, it has been reported that 1-hour plasma glucose concentration (1-h PG) during an OGTT can be a superior predictor of the risk of future T2DM, and people with elevated 1-h PG have a high rate of progression to T2DM even though they have normal glucose tolerance [51–53]. For early diagnosis and intervention for intermediate hyperglycemia and T2DM, the International Diabetes Federation recently suggested using the 1-h PG as an indicator [54]. This may indicate the importance of glucose effectiveness in the prediction of T2DM, since, based on our results, the hypoglycemic

effect within 1 h after glucose loading is mainly due to glucose effectiveness. However, given that this study was conducted in mice, we need to be careful when applying the results to humans because biological systems and responses may differ among different species.

Nevertheless, we believe this study lays important groundwork for the use of mathematical models in elucidating the detailed mechanisms of the hypoglycemic effects in both mouse and human studies. Our approach provides new perspectives for human studies and insights into the mechanisms underlying glucose effectiveness and insulin-dependent hypoglycemic effects.

## Methods

### Ethics statement

Animal experiments were approved by the Institutional Animal Care and Use Committee of Kyushu University and conducted according to the Kyushu University Animal Experiment Regulations.

### Animals

Male C57BL/6J mice at 12 and 4 weeks of age were purchased from Charles River Laboratories Japan, Inc. and used in the IVGTT (12 weeks of age) and hyperglycemic clamp test (4 weeks of age). Four-week-old mice were housed in Kyushu University's animal facility at 22 °C and with a 12-h light/dark cycle until they were used in the hyperglycemic clamp test. After 1 week of acclimatization, 5-week-old mice were divided into two groups. One group was fed a normal chow diet (CD) (D12450J, Research Diets) until 14 weeks of age, and the other group was fed a 60% high-fat diet (HFD) (D12492, Research Diets) until 10, 14, 18, 28, and 36 weeks of age. Grouping was performed randomly to avoid experimental bias. All mice were euthanized by cervical dislocation under deep isoflurane anesthesia. Isoflurane anesthesia was used for all procedures, and all efforts were made to minimize animal suffering, in accordance with the institutional guidelines approved by Kyushu University.

### IVGTT

After overnight fasting, 12-week-old mice were anesthetized with isoflurane and catheters were inserted into the jugular vein. The mice were divided into two groups: with or without somatostatin (Bachem AG, Bubendorf, Switzerland) dissolved in PBS containing 0.1% BSA. Somatostatin or PBS solution was infused (3 µg/kg/min) 30 minutes before a bolus glucose injection and infused at a constant rate using syringe pumps until the end of the test. A bolus glucose solution was infused for 3 minutes at a dose of 0.75 g/kg using a 15% glucose solution through the catheter. Blood samples were collected from the tail vein at 0, 1, 2, 3, 5, 10, 20, 30, 60, and 90 min to measure blood glucose and insulin levels.

### Hyperglycemic clamp test

After overnight fasting, the mice were anesthetized with isoflurane, and catheters were inserted into the jugular vein. During the hyperglycemic clamp test, we set the target blood glucose levels at 200 mg/dL higher than the fasting glucose levels and maintained the blood glucose levels at a constant by changing the administration rate using a 25% glucose solution as required. The initial glucose infusion rate, until reaching the target glucose levels (< 20 min), was also altered depending on the conditions. Blood glucose levels were measured at 0, 1, 2, 3, and 5 min during the first 5 min, every 5 min from 5 to 70 min, and every 10 min from 70 to 180 min from the tail vein. Blood samples were collected from the tail vein at 0, 1, 2, 3, 5, 10, 20, 30, 45, 60, 90, 120, 150, and 180 min to measure blood insulin levels. For 14-week-old chow-fed mice, the initial glucose infusion was performed at two rates using syringe pumps: slow-rate (0.15 µL/g/min) and fast-rate (4 µL/g/min) (the target blood glucose levels: approximately 320 mg/dL). For high-fat-fed mice, the initial glucose infusion rates differed based on the age of mice: 0.1 µL/g/min in 10, 14, and 18-week-old, and 0.13 µL/g/min in 28- and 36-week-old mice to reach the target glucose levels around 20 min. The target blood glucose levels were approximately 380, 420, 430, 400, and 380 mg/dL.

## Blood glucose and plasma insulin measurements

Blood glucose levels were determined using an AccuChek Aviva Nano Blood Glucose Meter (Roche Diagnostics). Blood insulin levels were measured by ELISA using a Levis Mouse Insulin Kit (Shibayagi, Gunma, Japan).

## Determination of glucose uptake in the liver and gastrocnemius muscles

The hyperglycemic clamp test was performed in 14-week-old chow-fed mice with the initial glucose infusion rate of 0.15 μL/g/min as described above. To measure 2-deoxyglucose (2-DG) uptake by the liver and the gastrocnemius muscles of the mice at 10, 20, 60, 120, and 180 min in the clamp test, 5 μmol of 2-DG was co-administered from 10 min before each time point. The livers and gastrocnemius muscles were obtained and immediately snap-frozen in liquid nitrogen after killing at each time point, and stored at −80 °C. The tissue samples were ground with Halt protease and phosphatase inhibitor cocktail (Thermo Fisher Scientific, Waltham, MA, USA) and dry ice four times for 20 s each using a blender (Wonder Blender, OSAKA CHEMICAL Co., Ltd, Osaka, Japan). The powdered tissues (10 mg of the livers and gastrocnemius) were homogenized in 500 μl 10 mM Tris–HCl (pH 8.0), heated at 95 °C for 15 min, and centrifuged at 17,800 g for 15 min at 4 °C. The resulting supernatants were assayed for 2-deoxyglucose 6-phosphate content using an enzymatic photometric assay with a 2-deoxyglucose uptake measurement kit (COSMO BIO Co., Ltd., Tokyo, Japan).

The amount of glucose absorbed by each organ was calculated as follows:

Amount of glucose absorbed by each organ (mg) = [Amount of glucose administered during 10 min (mg)] * [Amount of 2-DG uptake in each organ (nmol)]/ [Amount of 2-DG administered (nmol)].

The total amount of glucose in the circulating blood was calculated according to the blood glucose level (mg/dL) and individual circulating blood volume, which was calculated by multiplying body weight with circulating blood volume (72 mL/kg) [55].

## Developing a mathematical model

We developed the current model based on that proposed by Ohashi et al [38]. In the model, the variables $G$, $I$, and $stG$ represent blood glucose, insulin levels, and the transient storage of glucose, respectively. *Influx G* is infused with glucose and changes the infusion rate depending on the time required to maintain a constant blood glucose level. The fluxes in the model indicate mass transfer between the variables. *Flux 3* regulates insulin secretion depending on the effective glucose concentration and $k_3$ corresponds to the rate constant of insulin secretion. *Flux 4* represents insulin degradation and corresponds to insulin clearance. *Flux 2* corresponds to insulin-dependent glucose uptake. *Flux 1* corresponds to EGP and is suppressed by $I$. Glucagon promotes glycogenolysis and gluconeogenesis during hypoglycemia, thereby increasing hepatic glucose release and raising blood glucose levels. The effects of glucagon are incorporated into *flux 1*, which reflects these processes. Because it has been reported that glucose effectively suppresses EGP, we added $k_8$ to represent it. *Flux 5* corresponds to insulin-independent glucose uptake (glucose effectiveness), and $k_7$ represents the upper limit of glucose uptake by glucose effectiveness. *Flux 6* corresponds to insulin-dependent glucose consumption in $stG$.

The model is described in Fig 4A and follows:

$$dG/dt = flux\ 1 - flux\ 2 - flux\ 5 + Influx\ G = k_1/(1 + I + k_8{}^*G) - k_2{}^*G^*I - k_5{}^*G^*(k_7 - stG) + f(t)$$

$$dI/dt = flux\ 3 - flux\ 4 = k_3{}^*G - k_4{}^*I$$

$$dstG/dt = flux\ 5 - flux\ 6 = k_5{}^*G^*(k_7 - stG) - k_6{}^*stG^*I$$

The total amount of glucose uptake was calculated as follows:

$$dAG/dt = flux\,2 - flux\,1 + flux\,5 = k_2{}^*G^*I - k_1/(1 + I + k_8{}^*G) + k_5{}^*G^*(k_7 - stG)$$

## Parameter estimation

We used MATLAB R2016a and R2021b for the parameter estimation and simulation. The model parameters for each mouse were estimated to reproduce the normalized time course using a meta-evolutionary programming method to approach the neighborhood of the local minimum, followed by the application of the nonlinear least squares technique to reach the local minimum [38]. The parameters were estimated to minimize the objective function value, which was defined as the residual sum of the squares (RSS) between the data obtained using the hyperglycemic clamp test (blood glucose levels, blood insulin levels, and the total amount of glucose uptake) and the model trajectories. The RSS is given by the equation

$$RSS = \sum_{i=1}^{nG}([G(t_i) - Gsim(t_i)]/\sqrt{\sum_{i=1}^{nG}|G(t_i)|^2})^2 + \sum_{j=1}^{nI}([I(t_j) - Isim(t_j)]/\sqrt{\sum_{j=1}^{nI}|I(t_j)|^2})^2 ||$$

$$+ \sum_{i=1}^{nA}([AG(t_i) - AGsim(t_i)]/\sqrt{\sum_{i=1}^{nA}|AG(t_i)|^2})^2 ||$$

where $G(t_i)$, $I(t_j)$, and $AG(t_i)$ are the measured blood glucose and insulin levels, the calculated total amount of glucose uptake at t min, and the $i$-th/$j$-th time point ($i$ and $j$ are the indexes of the time points. $t_i$ = 0, 1, 2, 3, 5, 10, 15, 20, 25, 30, 35, 40, 45, 50, 55, 60, 65, 70, 80, 90, 100, 110, 120, 130, 140, 150, 160, 170, and 180 min. $t_j$ = 0, 1, 2, 3, 5, 10, 20, 30, 45, 60, 90, 120, 150, and 180 min). $nG$, $nI$, and $nA$ represent the total number of time points. $AG(t_i)$ is calculated as follows:

$$AG(t_i) = \sum_{i=1}^{ti}(f(t_i) * (t_i - t_{i-1}) - [G(t_i) - G(t_{i-1})]^* \text{ individual circulating blood volume}$$

$$t_0 = 0$$

$Gsim(t_i)$, $Isim(t_j)$, and $AGsim(t_i)$ are the simulated blood glucose and insulin levels, accumulated glucose uptake in the body at $t$ min, and the $i$-th/ $j$-th time point. $f(t_i)$ is the infused glucose at $t_i$ min. The initial blood glucose and insulin levels (at time 0) were defined as $G(0)$ and $I(0)$, respectively, and included as estimated parameters in the model.

## Statistical analysis

Data were analyzed using Microsoft® Office Excel and R software (ver. 4.1.2). Values are expressed as mean ± SE. Two-sided Welch's $t$ test and $p$-value adjustment by Benjamini-Hochberg method were used to assess the significance of differences between the means. Significant differences are indicated by $*p < 0.05$, $**p < 0.01$, $***p < 0.001$, and $****p < 0.0001$. The results of the parameters were expressed as means with first and third quartiles. The results smaller than $Q1 - 1.5IQR$ or greater than $Q3 + 1.5IQR$ are regarded as outliers [56] and excluded. For selected parameters ($k_2$, $k_3$, $k_5$ and $k_7$) that were hypothesized to exhibit a monotonic trend, Williams' test was applied to assess time-dependent changes. The

significance of the relationship between plasma insulin and infused glucose levels was assessed using Spearman's rank correlation coefficients. The relationship between $k_7$ and mouse weight was calculated using Spearman's rank correlation coefficient.

**Model deposit**

The model has been deposited in GitHub at https://github.com/Kubota001/Insulin-Glucose-model/commit/3a1053a7475499662dd41a6c1683b7a6452fe75d.

## Supporting information

**S1 Fig. Blood glucose levels decreased regardless of insulin secretion. Related to** Fig 1. Time-courses of blood glucose (A) and insulin levels (B) during the IVGTT with and without somatostatin in 12-week-old chow-fed mice. Orange and blue lines indicate the time courses with and without somatostatin, respectively. Results are expressed as mean±SE ($n$=3).
(PDF)

**S2 Fig. The time courses during the hyperglycemic clamp with 2-DG for the indicated time points. Related to** Fig 3. The time courses of blood glucose (left) and insulin (middle) levels, and the amount of infused glucose (right) during the hyperglycemic clamp with 2-DG for 10, 20, 60, and 120 minutes ($n$=3). All results are expressed as mean±SE. The hyperglycemic clamp test was performed in 14-week-old chow-fed mice with the initial glucose infusion rate of 0.15 µL/g/min and 5 µmol of 2-DG was co-infused from 10 min before each time point. The livers and gastrocnemius muscles were obtained to measure 2-DG uptake at indicated time point.
(PDF)

**S3 Fig. Experimental results versus mathematical model simulations in 14-week-old chow-fed mice and 10, 14, 18, 28, and 36-week-old HFD-fed mice. Related to Fig 6.** The time courses of blood glucose (left) and insulin (middle) levels, and the amount of infused glucose (right) during the hyperglycemic clamp for each mouse used in the developed models. Orange dots and blue lines indicate experimental and simulation results, respectively. # shows the number of individual mice.
(PDF)

**S1 Table. Estimated parameters and RSS for each mouse.** $G$ (0) and $I$ (0) are estimated initial blood glucose and insulin levels. "No" indicates the number of individual mice, which corresponds to the number in Fig 4B, 6, and S3 Fig.
(PDF)

**S1 Data. Blood glucose measurements of mice.** This file contains the underlying blood glucose data from all experiments presented in the manuscript.
(XLSX)

**S2 Data. Blood insulin measurements of mice.** This file contains the underlying blood insulin data from all experiments presented in the manuscript.
(XLSX)

**S3 Data. Body weight measurements of mice.** This file contains the underlying body weight data from all experiments presented in the manuscript.
(XLSX)

**S4 Data. Glucose doses administered to mice.** This file contains the underlying data of glucose doses administered at each time point in all experiments presented in the manuscript.
(XLSX)

## Acknowledgments

The authors thank Dr. Fumiko Matsuzaki (Kyushu University) and members of the Department of Medicine and Bioregulatory Science, Graduate School of Medical Sciences, Kyushu University for their helpful comments regarding this project.

## Author contributions

**Conceptualization:** Hiroyuki Kubota.

**Data curation:** Fusako Kojima.

**Formal analysis:** Fusako Kojima.

**Funding acquisition:** Hiroyuki Kubota.

**Investigation:** Fusako Kojima.

**Methodology:** Fusako Kojima, So Morishita, Shinsuke Uda.

**Project administration:** Hiroyuki Kubota.

**Resources:** Fusako Kojima, So Morishita, Noriko Yutsudo.

**Supervision:** Hiroyuki Kubota, Shinsuke Uda, Yoshihiro Ogawa.

**Validation:** Fusako Kojima.

**Visualization:** Fusako Kojima, So Morishita.

**Writing – original draft:** Fusako Kojima, Hiroyuki Kubota.

**Writing – review & editing:** Fusako Kojima, Hiroyuki Kubota, So Morishita, Shinsuke Uda, Noriko Yutsudo, Yoshihiro Ogawa.

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
