## [Decision Letter · Decision Letter 0]

1 Sep 2025

PONE-D-25-10318Evaluation of the insulin-dependent and –independent hypoglycemic effects and understanding their breakdown in the progression of obesity using micePLOS ONE

Dear Dr. Kubota,

Thank you for submitting your manuscript to PLOS ONE. After careful consideration, we feel that it has merit but does not fully meet PLOS ONE’s publication criteria as it currently stands. Therefore, we invite you to submit a revised version of the manuscript that addresses the points raised during the review process.

The manuscript has now been critically evaluated by two exeprts in the field. There is a clear consent that the paper is suitable for publication after major and minor points raised by the reviewer have been adequately addressed. 

For details please refer to the reviewers comments below. 

We look forward to receiving your revised manuscript.

Kind regards,

Stefan Wölfl, Ph.D.

Academic Editor

PLOS ONE

**Please note that the academic editor has acted as a reviewer for this manuscript, and you will find the comments below, under Reviewer 2.**

Journal Requirements:

“The Japan Society for the Promotion of Science  KAKENHI, grant number JP20H03237. the Japan Science and Technology Agency (JST) Moonshot R&D, grant number JPMJMS2022-8”

4. Please note that funding information should not appear in the Acknowledgments section or other areas of your manuscript. We will only publish funding information present in the Funding Statement section of the online submission form. Please remove any funding-related text from the manuscript. 

6. Please amend either the title on the online submission form (via Edit Submission) or the title in the manuscript so that they are identical.

7. We note that you have included the phrase “data not shown” in your manuscript. Unfortunately, this does not meet our data sharing requirements. PLOS does not permit references to inaccessible data. We require that authors provide all relevant data within the paper, Supporting Information files, or in an acceptable, public repository. Please add a citation to support this phrase or upload the data that corresponds with these findings to a stable repository (such as Figshare or Dryad) and provide and URLs, DOIs, or accession numbers that may be used to access these data. Or, if the data are not a core part of the research being presented in your study, we ask that you remove the phrase that refers to these data.

8. Please include a separate caption for each figure in your manuscript.

**Additional Editor Comments:**

Thank you for submitting your manuscript for publication in PLoS one. The work is very impressing and I am happy that we will be able to publish your manuscript after the points raised in the reviewers comments have been adequately addressed. Please excuse the long initial reviewing process, but it was very difficult to find adequate reviewers for your manuscript, since it comprises several parts of expertise in physiology, in vivo monitoring of glucose levels and mathematical modelling.

Reviewers' comments:

Reviewer's Responses to Questions

**Comments to the Author**

1. Is the manuscript technically sound, and do the data support the conclusions?

Reviewer #1: Yes

2. Has the statistical analysis been performed appropriately and rigorously? 

Reviewer #1: Yes

3. Have the authors made all data underlying the findings in their manuscript fully available?

Reviewer #1: No

4. Is the manuscript presented in an intelligible fashion and written in standard English?

Reviewer #1: Yes

5. Review Comments to the Author

Reviewer #1: This paper dissects by means of direct measurements during mainly hyperglycemic clamp (HGC) but also IVGTT in mice the contributions of glucose effectiveness (SG) and insulin sensitivity (SI). This is combined with mathematical modeling. The model addresses some of the same phenomena as the classic Minimal Model but includes more mechanistic processes. It concludes that SG dominates during the first 60 min after an intravenous glucose load, whereas SI grows in importance from 60 - 180 min. Changes during the development of obesity are tracked in mice on a high fat diet by fitting the model to individual mice and plotting model parameters vs. time.

The results are plausible, and I have mainly minor comments to improve and clarify the presentation.

Major Comments:

1. L. 92 - 93, during a 90-min IVGTT application of somatostatin to suppress insulin secretion had no effect on rate of glucose uptake (Fig. S1). Would insulin have had more effect if the test had been continued beyond 90 min? I would like to see a comparison with two classic studies in people with T1D with and without exogenous insulin, which found that MinMod overestimated the magnitude of SG during IVGTT: PMID: 8013753 and PMID: 9843746.

2. L. 167, "previous models ... cannot reproduce the transient glucose effectiveness ... because onlu constant glucose effectiveness is observed under a constant blood glucose": This explanation seems a bit too simple. Flux 5 in the new model also depends only on G, and the coefficients k5 and k7 are constant. The only factor I can see that might account for the transient peak of glucose effectiveness in Fig. 4C is (k7 - stG), which limits flux 5 as stG grows. Please comment.

3. L. 706 - 708: The model is fit to three inputs, G, I and AG. I am concerned that this is not sufficient to identify eight parameters. Please justify.

4. Please post the computer code for the model in a public repository.

Minor Comments:

1. L. 64, "glucose effectiveness is responsible for ... 45 - 65% of glucose disposal": What is the context? During an IVGTT or other test?

2. L. 120: "little hypoglycemic effects" should be "little hypoglycemic effect"."

3. L. 138: "Therefore"; "This explains why" would be better

4. L. 176, "EGP is suppressed by glucose effectiveness": Do you mean that glucose uptake into the liver suppresses EGP? SG is a process, not a molecular species.

5. L. 184, Figs. 3 and 4, "estimates are comparable": The numbers are on very different scales, and there are no units for glucose in Fig. 4. Also, G is in mg/dl in some figures and mM in others. In Fig. 5B, the glucose unit is mg. Please correct and standardize.

6. L. 187: The phrase beginning "While" is a sentence fragment.

7. L. 200, "time-dependent glucose uptake": Please clarify here and elsewhere whether "time-dependent" means over minutes or weeks.

8. L. 222, "parameter k_5 ... appeared to decrease": Can you do a statistical test to establish whether it did decrease?

9. L. 232, "the model captured changes in glucose homeostasis": It certainly predicted changes but how do we know they are correct?

10. L. 240, "and not effected by blood insulin levels": redundant, please delete.

11. L. 326, "a high-fat diet reduces hepatic NAD+ to decrease hepatic Sirt2 activity": "to decrease" implies purpose. Better to say "has the effect of decreasing".

12. L. 326, "This study": Which study?

13. L. 338: 'The question of "when" ... hypoglycemic effects occur': No need for "'s.

14. Figure 4 legend: Define "CD".

15. L. 749: "old chow" should be "old chow".

Reviewer #2

In the presented manuscript "Evaluation of the insulin-dependent and –independent hypoglycemic effects and understanding their breakdown in the progression of obesity using mice " the authors present a impressive amount of data measuring blood glucose levels and a detailed analysis of analyse the effect of glucose uptake in various tissues in an obese mouse model. The methods are described well and the results are adequately presented. There is however a significant omission which needs to be addressed before the mansucript can be published. In their model the authors use somatostatin administration as model for "insulin"-independent glucose only response to increased glucose levels. The manuscript however lacks information that somatostatin not only blocks insulin secretion but also glucagon secretion in the pancreatic island cells. This is very essential for the interpretation of the results because glucagon plays a central role in adjusting/controling glucose uptake and release in the liver and may also influence glucose uptake in muscle cells. Therefore, it is very likely that inhibition of glucagon secretion contributes to the "glucose"-only effect. This needs to be discussed in the paper and the mathematical modeling should be adjsuted for this further parameter. Unfortunately, there are even more physiological regulators linked with peripheral tissue homeostatsis, but I agree that these can remain as a "black box" contributing in qn equal manor to blood glucose levers in the model presented.

recommendation: major revision

6. PLOS authors have the option to publish the peer review history of their article (what does this mean? ). If published, this will include your full peer review and any attached files.

**Do you want your identity to be public for this peer review?** For information about this choice, including consent withdrawal, please see our Privacy Policy .

Reviewer #1: No

---

## [Author Response · Author response to Decision Letter 1]

15 Oct 2025

Response to Reviewer #1

Reviewer #1: This paper dissects by means of direct measurements during mainly hyperglycemic clamp (HGC) but also IVGTT in mice the contributions of glucose effectiveness (SG) and insulin sensitivity (SI). This is combined with mathematical modeling. The model addresses some of the same phenomena as the classic Minimal Model but includes more mechanistic processes. It concludes that SG dominates during the first 60 min after an intravenous glucose load, whereas SI grows in importance from 60 - 180 min. Changes during the development of obesity are tracked in mice on a high fat diet by fitting the model to individual mice and plotting model parameters vs. time.

The results are plausible, and I have mainly minor comments to improve and clarify the presentation.

We thank the reviewer for the thorough reading of our manuscript and for providing suggestions that helped us to improve its quality. We have carefully revised our manuscript in accordance with the reviewer’s comments and suggestions. We believe that these revisions process greatly added insights and clarifications to our findings, thereby enhancing the overall quality of our manuscript. We would like to express our sincere appreciation once again for the opportunity to improve our manuscript, and we thank you for sparing your valuable time to review our responses. Below are point-by-point responses to reviewer comments:

Note that the editor’s comments have been incorporated to revise the document formatting. As a result, the placement of figure legends and the presentation of figures and references differ from the originally submitted version. Revisions addressing the reviewers’ comments are highlighted in yellow.

Major Comments:

1. L. 92 - 93, during a 90-min IVGTT application of somatostatin to suppress insulin secretion had no effect on rate of glucose uptake (Fig. S1). Would insulin have had more effect if the test had been continued beyond 90 min? I would like to see a comparison with two classic studies in people with T1D with and without exogenous insulin, which found that MinMod overestimated the magnitude of SG during IVGTT: PMID: 8013753 and PMID: 9843746.

Our response: Thank you for your comments reinforcing our conclusion regarding the initial insulin-independent glucose uptake and the subsequent insulin-dependent glucose uptake. We expect that the effect of insulin would have been enhanced beyond 90 minutes. At the time we performed the 90-minute IVGTT experiment in the presence of somatostatin (Fig. S1), we had not yet reached the conclusion of this study. Therefore, we did not conduct experiments beyond 90 minutes. As the Editor requested a response within approximately one month, we have addressed this point by adding discussions in the main text, citing relevant papers (Ref. 1: Sjöstrand et al., Diabetes, 2002; Ref. 2: Mossberg & Taegtmeyer, J Nucl Med, 1992, lines 94-98, 342-357, and 415-418). However, anticipating that the reviewer might not be satisfied with this response alone, we have now submitted an ethics review application for a longer IVGTT experiment under somatostatin conditions extending beyond 90 minutes. This process, including the experiment itself, is expected to take at least three to four months. Should the above discussion be deemed insufficient, we will proceed with additional experiments.

We have also added a comparison with the two classic studies you pointed out (Ref. 3: PMID: 8013753 and Ref. 4:PMID: 9843746) in the revised manuscript (lines 342-357 and 361-364).

2. L. 167, "previous models ... cannot reproduce the transient glucose effectiveness ... because onlu constant glucose effectiveness is observed under a constant blood glucose": This explanation seems a bit too simple. Flux 5 in the new model also depends only on G, and the coefficients k5 and k7 are constant. The only factor I can see that might account for the transient peak of glucose effectiveness in Fig. 4C is (k7 - stG), which limits flux 5 as stG grows. Please comment.

Our response: We apologize for the insufficient explanation. As the reviewer commented, the flux5 of the new model depends on G, and k5 and k7 are constant. To clarify this point, we have revised the main text (lines 199–203 and 208–210).

3. L. 706 - 708: The model is fit to three inputs, G, I and AG. I am concerned that this is not sufficient to identify eight parameters. Please justify.

Our response: We believe that Reviewer #1 is concerned about a potentially degenerate solution for the eight parameters estimated using the least squares method. Evaluating parameter identifiability in ODE fitting is challenging, particularly in practical systems biology applications. In the case of linear systems of equations, the sample-size-to-parameter ratio is considered to serve as an indicator for assessing potential degeneracy. Although our fitting problem is nonlinear due to the simultaneous ordinary differential equations (ODEs), this ratio can still serve as a rough indicator for assessing potential degeneracy, even in nonlinear contexts. In our analysis, a total of 10 parameters—8 main parameters plus G (0) (blood glucose at 0 minutes) and I (0) (blood insulin at 0 minutes)— were fitted to a dataset of 29 samples, resulting in a sample-size-to-parameter ratio of 2.9. By comparison, the ratios reported by Shiang et al. (Ref. 5) and Brenner et al. (Ref. 6), both of which developed mathematical models of glucose-insulin regulation, are 1 and 3.8, respectively. Therefore, we believe that sufficient information has been obtained given our parameter and data sizes.

4. Please post the computer code for the model in a public repository.

Our response: Thank you for your comments. In accordance with the suggestion, the model has been deposited in GitHub at “https://github.com/Kubota001/Insulin-Glucose-model/commit/3a1053a7475499662dd41a6c1683b7a6452fe75d”.

Minor Comments:

1. L. 64, "glucose effectiveness is responsible for ... 45 - 65% of glucose disposal": What is the context? During an IVGTT or other test?

Our response: We apologize for the insufficient explanation. This result comes from a minimal model analysis using IVGTT and glucose clamp experiments (Refs. 7-10). To clarify this point, we have revised the text (lines 57–60).

2. L. 120: "little hypoglycemic effects" should be "little hypoglycemic effect"."

Our response: Thank you for pointing this out. We have corrected this error (lines 121-122).

3. L. 138: "Therefore"; "This explains why" would be better

Our response: Thank you for pointing this out. We have corrected this error (line 151).

4. L. 176, "EGP is suppressed by glucose effectiveness": Do you mean that glucose uptake into the liver suppresses EGP? SG is a process, not a molecular species.

Our response: Thank you for pointing this out. It is reported that EGP is suppressed not only by insulin but also by glucose (Refs. 11-13). Since in these studies, they described that “'Glucose effectiveness suppresses endogenous glucose production and stimulates glucose uptake,” we used this expression. However, as the reviewer commented, SG is a process, not a molecular species. Therefore, we revised the expression (lines 211-216).

5. L. 184, Figs. 3 and 4, "estimates are comparable": The numbers are on very different scales, and there are no units for glucose in Fig. 4. Also, G is in mg/dl in some figures and mM in others. In Fig. 5B, the glucose unit is mg. Please correct and standardize.

Our response: Thank you for your advice. In accordance with your suggestions and corrections, we have revised Fig. 4B as follows, as well as all figures in the Supporting information. We have also confirmed “unit” in all figures.

6. L. 187: The phrase beginning "While" is a sentence fragment.

Our response: Thank you for pointing this out. In accordance with your advice, we have revised it

(line 234).

7. L. 200, "time-dependent glucose uptake": Please clarify here and elsewhere whether "time-dependent" means over minutes or weeks.

Our response: We thank the reviewer for their suggestion. In this study, we describe the two-stage glucose uptake process—an initial phase (within 60 min) and a late (after 60 min) phase—as “time-dependent glucose uptake.” To clarify this point, we have revised the sentence (lines 235-237).

8. L. 222, "parameter k_5 ... appeared to decrease": Can you do a statistical test to establish whether it did decrease?

Our response: We thank the reviewer for their valuable suggestion, which strengthens our results. Following the suggestion, we performed the Williams’ test, which assesses whether the data exhibit monotonic trends, and found that k5 significantly decreased during obesity progression under HFD feeding. In addition, we further examined whether other changed parameters (k2, k3, andd k7) were also age-dependently altered, and found that they likewise exhibited age-dependent trends. These results and corresponding discussions have been added to the text (lines 267-292 and 610-611).

9. L. 232, "the model captured changes in glucose homeostasis": It certainly predicted changes but how do we know they are correct?

Our response: We thank the reviewer for their comment. As the reviewer correctly noted, this is a prediction. Therefore, we have softened the expression (lines 294-295).

10. L. 240, "and not effected by blood insulin levels": redundant, please delete.

Our response: We thank the reviewer for their suggestion. Accordance with their advice, we have deleted the sentence (line 324).

11. L. 326, "a high-fat diet reduces hepatic NAD+ to decrease hepatic Sirt2 activity": "to decrease" implies purpose. Better to say "has the effect of decreasing".

12. L. 326, "This study": Which study?

Our response: Thank you for pointing these out. We have revised them (lines 433-436).

13. L. 338: 'The question of "when" ... hypoglycemic effects occur': No need for "'s.

Our response: Thank you for pointing this out. We have revised it accordingly (line 445).

14. Figure 4 legend: Define "CD".

Our response: Thank you for pointing this out. Accordance with the advice, we have added a definition of “CD” on the figure legend (lines 220-224).

15. L. 749: "old chow" should be "old chow".

Our response: Thank you for pointing out the typographical error. We have corrected it (lines 784-785).

Response to Reviewer #2

In the presented manuscript "Evaluation of the insulin-dependent and –independent hypoglycemic effects and understanding their breakdown in the progression of obesity using mice " the authors present a impressive amount of data measuring blood glucose levels and a detailed analysis of analyse the effect of glucose uptake in various tissues in an obese mouse model. The methods are described well and the results are adequately presented. There is however a significant omission which needs to be addressed before the mansucript can be published.

We thank the reviewer for the thorough reading of our manuscript. We are pleased to hear that you found our methodology to be clearly described and the results of our study to be adequately demonstrated.

Note that the editor’s comments have been incorporated to revise the document formatting. As a result, the placement of figure legends and the presentation of figures and references differ from the originally submitted version. Revisions addressing the reviewers’ comments are highlighted in yellow.

In their model the authors use somatostatin administration as model for "insulin"-independent glucose only response to increased glucose levels. The manuscript however lacks information that somatostatin not only blocks insulin secretion but also glucagon secretion in the pancreatic island cells. This is very essential for the interpretation of the results because glucagon plays a central role in adjusting/controling glucose uptake and release in the liver and may also influence glucose uptake in muscle cells. Therefore, it is very likely that inhibition of glucagon secretion contributes to the "glucose"-only effect. This needs to be discussed in the paper and the mathematical modeling should be adjsuted for this further parameter.

Unfortunately, there are even more physiological regulators linked with peripheral tissue homeostatsis, but I agree that these can remain as a "black box" contributing in qn equal manor to blood glucose levers in the model presented.

Our response: We sincerely appreciate your valuable comment. We agree that glucagon is a crucial hormone in blood glucose homeostasis. In the IVGTT (Fig S1) that initiated this study, experiments were conducted with or without somatostatin. However, all subsequent experiments were conducted without somatostatin. Therefore, in this study, excluding Fig S1, there is no influence of glucose regulation mediated by somatostatin-induced suppression of glucagon secretion. To clarify this point, we have added a description discussing the role of glucagon under our experimental conditions, shown in Fig S1 and specifying those conditions (lines 94-98).

Nevertheless, the role of glucagon remains important in the regulation of blood glucose levels. On the other hand, insulin has been reported to suppress glucagon secretion (Refs. 14-16). Indeed, in glucose clamp, IVGTT, and OGTT experiments, insulin and glucagon exhibit opposing dynamics (Refs. 17-18). This relationship has been incorporated into the current model (flux1), as in other previous models, including the minimal model and our previous model. In other words, although not explicitly stated, the effect of glucagon is implicitly included in these models. Furthermore, while these effects primarily act on the liver, as the reviewer noted, glucagon may also have effects on muscle. However, the studies we reviewed demonstrated that an increase in glucagon does not affect glucose uptake in skeletal muscles (Refs. 19-20); therefore, the effects of glucagon on muscle remain controversial. In addition, our model was not developed using experiments with somatostatin (Fig S1). Therefore, incorporating the effects of glucagon into our current model is considered premature at this stage. Nevertheless, as research progresses, it may become possible to extend our model to examine the effects of glucagon on muscle, as well as the influence of other hormones with hyperglycemic effects, such as those mentioned by the reviewer (e.g., blood glucose regulation under stress conditions). Therefore, we have added a discussion addressing this point as a further issue (lines 211-216 and 457-463).

References

1. Sjostrand M, Gudbjornsdottir S, Holmang A, Lonn L, Strindberg L, Lonnroth P. Delayed transcapillary transport of insulin to muscle interstitial fluid in obese subjects. Diabetes. 2002;51(9):2742-8.

2. Mossberg KA, Taegtmeyer H. Time course of skeletal muscle glucose uptake during euglycemic hyperinsulinemia in the anesthetized rabbit: a fluorine-18-2-deoxy-2-fluoro-D-glucose study. J Nucl Med. 1992;33(8):1523-9.

3. Quon MJ, Cochran C, Taylor SI, Eastman RC. Non-insulin-mediated glucose disappearance in subjects with IDDM. Discordance between experimental results and minimal model analysis. Diabetes. 1994;43(7):890-6.

4. Cobelli C, Bettini F, Caumo A, Quon MJ. Overestimation of minimal model glucose effectiveness in presence of insulin response is due to undermodeling. Am J Physiol. 1998;275(6): E1031-6.

5. Shiang KD, Kandeel F. A computational model of the human glucose-insulin regulatory system. J Biomed Res. 2010;24(5):347-64.

6. Brenner M, Abadi SEM, Balouchzadeh R, Lee HF, Ko HS, Johns M, et al. Estimation of insulin secretion, glucose uptake by tissues, and liver handling of glucose using a mathematical model of glucose-insulin homeostasis in lean and obese mice. Heliyon. 2017;3(6): e00310.

7. Henriksen JE, Alford F, Handberg A, Vaag A, Ward GM, Kalfas A, et al. Increased glucose effectiveness in normoglycemic but insulin-resistant relatives of patients with non-insulin-dependent diabetes mellitus. A novel compensatory mechanism. J Clin Invest. 1994;94(3):1196-204.

8. Gottesman I, Mandarino L, Gerich J. Estimation and kinetic analysis of insulin-independent glucose uptake in human subjects. Am J Physiol. 1

---

## [Decision Letter · Decision Letter 1]

12 Nov 2025

Evaluation of the insulin-dependent and –independent hypoglycemic effects and understanding their breakdown in the progression of obesity using mice

PONE-D-25-10318R1

Dear Dr. Kubota,

We’re pleased to inform you that your manuscript has been judged scientifically suitable for publication and will be formally accepted for publication once it meets all outstanding technical requirements.

Kind regards,

Stefan Wölfl, Ph.D.

Academic Editor

PLOS ONE

Additional Editor Comments (optional):

The author have carefully revised their work and addressed most points raised by the reviewers and significantly improved the manuscript. Although some points in particular regarding the crosstalk between different physiological signals, the manuscript has been carefully rewritten and significantly improved.

Reviewers' comments:

Reviewer's Responses to Questions

**Comments to the Author**

1. If the authors have adequately addressed your comments raised in a previous round of review and you feel that this manuscript is now acceptable for publication, you may indicate that here to bypass the “Comments to the Author” section, enter your conflict of interest statement in the “Confidential to Editor” section, and submit your "Accept" recommendation.

Reviewer #1: All comments have been addressed

2. Is the manuscript technically sound, and do the data support the conclusions?

Reviewer #1: Yes

3. Has the statistical analysis been performed appropriately and rigorously? 

Reviewer #1: Yes

4. Have the authors made all data underlying the findings in their manuscript fully available?

Reviewer #1: Yes

5. Is the manuscript presented in an intelligible fashion and written in standard English?

Reviewer #1: Yes

6. Review Comments to the Author

Reviewer #1: The authors have carefully and systematically responded to the critiques of the first review, clarifying parts that were ambiguous and adding statistical analysis for trends in the model parameters. I have no further objections.

7. PLOS authors have the option to publish the peer review history of their article (what does this mean? ). If published, this will include your full peer review and any attached files.

**Do you want your identity to be public for this peer review?** For information about this choice, including consent withdrawal, please see our Privacy Policy .

Reviewer #1: **Yes: ** Arthur Sherman

---

## [Editor Report · Acceptance letter]

PONE-D-25-10318R1

PLOS One

Dear Dr. Kubota,

I'm pleased to inform you that your manuscript has been deemed suitable for publication in PLOS One. Congratulations! Your manuscript is now being handed over to our production team.

Kind regards,

on behalf of

Prof. Dr. Stefan Wölfl

Academic Editor

PLOS One